# Nutrition Strategies to Promote Sleep in Elite Athletes: A Scoping Review

**DOI:** 10.3390/sports13100342

**Published:** 2025-10-02

**Authors:** Gavin Rackard, Sharon M. Madigan, James Connolly, Laura Keaver, Lisa Ryan, Rónán Doherty

**Affiliations:** 1Department of Tourism & Sport, Atlantic Technological University, Letterkenny Campus, Port Road, Letterkenny, F92 FC93 Co. Donegal, Ireland; ronan.doherty@atu.ie; 2Sport Ireland Institute, National Sports Campus, Abbotstown, D15 Y52H Dublin, Ireland; smadigan@instituteofsport.ie; 3School of Computing, Engineering and Intelligent Systems, Ulster University, Northland Rd., Londonderry BT48 7JL, UK; jp.connolly@ulster.ac.uk; 4Department of Health and Nutritional Science, Atlantic Technological University, Ash Lane, F91 YW50 Co. Sligo, Ireland; laura.keaver@atu.ie; 5Department of Sport, Exercise and Nutrition, Atlantic Technological University, Galway City, Old Dublin Road, H91 T8NW Co. Galway, Ireland; lisa.ryan@atu.ie

**Keywords:** nutrition, sleep, elite athletes, NUQUEST, P2P Matrix, kiwifruit, tart cherry juice, carbohydrate, protein

## Abstract

Background/Objectives: Sleep is pivotal for recovery, immunity, and energy restoration; however, sleep problems exist in elite athletes. Nutrition and supplementation strategies can play both a positive and negative role in sleep quality and quantity. Elite athletes experience unique psychological and physiological demands above non-elite athletes and may require different nutrition strategies to promote sleep. Nutrient interventions and their effect on sleep in elite athletes is an emerging area, with further research warranted. Methods: Preferred Reporting Items for Systematic Reviews and Meta-Analyses (PRISMA) extension for Scoping Reviews and Joanna Brigg’s Institute Reviewer’s Manual for Scoping Reviews were utilised to assess the available evidence on nutrition strategies used to promote sleep in elite athlete cohorts, and we tried to identify the interventions that could be best researched in the future. NUtrition QUality Evaluation Strengthening Tools (NUQUEST) was used to enhance rigour and assess risk of bias in studies. The Paper to Podium (P2P) Matrix was used to offer practitioners practical recommendations. Results: 12 studies met the inclusion criteria for nutrition interventions or exposures to promote sleep in elite athletes. The median participant group size was 19 and study designs were considered together to ascertain potential sleep promoting strategies. Kiwifruit, Tart Cherry Juice and high dairy intake, limited to females, have demonstrated the highest potential to promote sleep in elite athletes, despite limited sample sizes. A-lactalbumin, carbohydrate pre-bed, casein, tryptophan, probiotic and meeting energy demands showed varying results on sleep quality in elite athletes. Conclusions: Kiwifruit, Tart Cherry Juice and dairy consumption offer potential nutritional interventions to promote sleep in elite athletic populations, while protein-based interventions may have a ceiling effect on sleep quality when elite athletes are already consuming >2.5 g·kg^−1^ body mass (BM) or are already meeting their sleep duration needs.

## 1. Introduction

Sleep is essential for the recovery, immunity and performance of athletes [1]. An increasing body of evidence highlights the sleep requirements of athletic populations [1,2,3,4,5], with research examining the interaction between nutrition and sleep in athletes being increasingly investigated in recent years [3,4,6,7,8,9].

The human sleep cycle consists of two phases: rapid eye movement (REM) and non-rapid eye movement (NREM) sleep, with NREM further divided into three stages (N1–N3) [10]. The body typically cycles through these stages 4–6 times nightly, with each cycle lasting approximately 90 min [4,10]. N1, the lightest sleep stage, features theta waves, while N2, the most common stage, includes sleep spindles and K-complexes, which aid memory consolidation. N3, or slow-wave sleep, is the deepest stage, crucial for physical repair and immune function [11]. REM sleep, accounting for 25% of total sleep, is characterised by vivid dreaming, muscle atonia and increased brain activity resembling wakefulness [10]. Sleep is regulated by the circadian rhythm, controlled by the suprachiasmatic nuclei, which influence hormone release, including melatonin and norepinephrine [10]. Gamma-aminobutyric acid (GABA) plays a key role in sleep induction by inhibiting wake-promoting brain regions through GABA-A receptor activation [10,12].

The American Academy of Sleep Medicine and Sleep Research Society recommend that the average adult should sleep 7 or more hours per night regularly to promote optimal health [13]. The National Sleep Foundation concurs with the recommendation of 7 to 9 h of sleep for adults and 7 to 8 h of sleep for older adults [13]. Individual needs vary, but consistently sleeping less than 7 h negatively impacts immunity, the endocrine system, cognitive function, and cardiovascular disease risk [13]. Short sleep duration (<7 h) is associated with coronary heart disease (CHD) and stroke, as well as increased risk of CHD mortality [14].

Sleep is a key recovery strategy in sports [15], yet elite athletes often experience reduced total sleep time (TST), total time in bed (TTB), and sleep quality after late-night games [16,17], and may spend longer TTB but have lower sleep efficiency (SE) than non-athletes, resulting in similar TST [5]. Additional factors such as nerves, adrenaline, caffeine, and unfamiliar environments can also impair sleep in this cohort [3,16]. Poor sleep can occur before major competitions due to performance anxiety or during regular training due to early sessions and performance-related stress [4]. Sargent [18] found that elite athletes report needing 8.3 ± 0.9 h of sleep but only obtain 6.7 ± 0.8 h, with 71% sleeping over an hour less than necessary, with just 3% meeting their requirement. Optimal sleep onset is around 22:00–22:30, with waking at 09:00–09:30 for an elite athletic population. Team-sport athletes (6.9 h) sleep more than individual-sport athletes (6.4 h), and females generally fall asleep earlier. Given recovery demands, 8.3 h of sleep per night is recommended for elite athletes.

As a result of elite athletes typically obtaining less sleep than these recommended benchmarks [18] and being particularly at risk of sleep inadequacies [1], their hormonal balance can be disrupted, recovery processes impaired, and they are at an increased risk of illness [18,19]. These sleep inadequacies are characterised by habitual sleep durations <7 h [20], long sleep onset latency (SOL) [21], sleep dissatisfaction, unrefreshing sleep [21], daytime sleepiness and daytime fatigue [20]. Studies reporting global sleep quality show that 50–78% of elite athletes experience sleep disturbance and 22–26% suffer highly disturbed sleep [1]. As a result of this, chronic poor sleep can be detrimental to the recovery of elite athletes by stunting muscle growth and repair because of the impaired release of growth hormone, increased inflammation, glycogen restoration, psychological well-being, and moderating hormone balance, particularly cortisol and testosterone [3,16]

From a performance perspective, this poor sleep can impact concentration levels, reaction time, energy levels during competition, hormonal balance, and injury risk [3]. Longer sleep durations can, however, correlate positively with finishing place in a multiday netball competition [15], where sleep was objectively evaluated in 42 netball players during a tournament, finding that higher-ranked teams slept longer, had greater TTB, and reported better subjective sleep ratings. Similarly, Leeder [5] investigated sleep in Olympic athletes (*n* = 47) using actigraphy, finding poorer sleep characteristics than in non-athletic controls (*n* = 20), with males exhibiting poorer outcomes than females.

In addition to the increased risk of clinically poor sleep, elite athletes face unique external stressors beyond those experienced by non-elite athletes, including greater training volume, shifting schedules, contracts, and travel demands [22,23]. Psychological differences include higher self-efficacy and a present-focused time perspective, while physiological differences involve superior VO_2_ max, anaerobic thresholds, and running economy in endurance sports [24,25]. Recognising these psychological and physiological distinctions is important for this scoping review, as sleep behaviours and the impact of nutrition interventions may differ between elite and non-elite cohorts. Due to the previous lack of a clear definition of “elite athletes” in research, Swann [22] developed a taxonomy that was further refined by McKay [23] to classify and report levels of elite athlete cohorts. McKay [23] introduced a six-tiered Participant Classification Framework to standardise categorisation and improve study reliability and comparability. To summarise, elite athletes compete at a highly competitive standard, typically national or international, with demonstrated success and significant experience. Their status is best viewed as a continuum based on multiple weighted factors, including competition level, competitive achievements, years of experience, and the competitiveness of their sport both locally and globally, providing a clear, objective framework for consistent classification [22,23].

A nutrient-dense diet supports sleep and recovery [9,26,27], with emerging evidence suggesting that nutrition interventions enhance sleep quality [28]. Optimising sleep is crucial for athletes, due to its influence on recovery, performance, and overall health. Several nutritional interventions, such as high GI carbohydrate intake, tryptophan supplementation, kiwifruit, tart cherry juice, zinc, and magnesium, have been explored to enhance sleep quality and duration in athletes. The potential for nutrition to improve sleep is clear; however, the lack of intervention studies specifically for elite athletes [3,4,6,9,29], and this scoping review justified the aim to offer practitioners clarity on context-specific nutrition interventions. Walsh [1] recommended that researchers collaborate and use more consistent methods to improve evidence quality and inform practice within sleep research. Therefore, a scoping review was conducted to systematically map the existing literature, assess the practical applicability of nutrition interventions and exposures using the P2P Matrix, and identify practical nutrition recommendations that influence sleep quality and quantity in elite athletes.

## 2. Materials & Methods

### 2.1. Protocol and Registration

A scoping review protocol compliant with PRISMA extension for Scoping Reviews [30] and Joanna Brigg’s Institute Reviewer’s Manual for Scoping Reviews [31] was developed. It followed the PRISMA-ScR 22-point checklist [30] Appendix A to help synthesize evidence and assess the scope of the literature. This review was registered on 1 July 2025, with Open Sciences Framework (https://doi.org/10.17605/OSF.IO/KUNWZ accessed on 4 December 2024).

### 2.2. Eligibility Criteria

This scoping review was guided by both the PICO (Population, Intervention, Comparator, Outcome) and PECO (Population, Exposure, Comparator, Outcome) frameworks to capture the full range of study designs relevant to the review question. The PICO model was applied to identify and include intervention-based studies (e.g., randomized controlled trials, controlled trials) that examined the impact of nutrition and supplement interventions on sleep outcomes in elite athletes. The PECO model was used to guide inclusion of observational studies (e.g., cohort, cross-sectional, case–control studies) where nutritional intake or dietary patterns were considered as exposures potentially associated with sleep outcomes. This dual-framework approach allowed for a comprehensive capture of both interventional and associative evidence related to the effects of nutrition on sleep in elite athletic populations.

To be eligible for inclusion in this scoping review, a study had to be conducted in elite athletes, as defined by the six-tiered Participant Classification Framework [22,23], and predefined in the introduction, ≥18 years old, free of any medically diagnosed health conditions, eating disorders (e.g., anorexia nervosa, bulimia nervosa), mental health conditions, diabetes, cancer, and cardiovascular disease.

Eligible studies reported that either (i) an intervention that modified components of diet (e.g., energy intake, macronutrient intake, intake of a specific food/beverage), or (ii) exposure to a dietary pattern and its effect on sleep (e.g., sleep duration, sleep efficiency, sleep timing). For this review, randomised controlled trials, randomised and non-randomised crossover studies, observational studies, and pre-post studies conducted in free-living, laboratory, and mixed settings were included. Reviews, meta-analyses, case studies, editorials, and conference abstracts were excluded.

Studies published in English and from any date in peer-reviewed journals were eligible for inclusion.

### 2.3. Search Strategy and Database Selection

The initial search strategy was developed in consultation with the authors and tested in the PubMed database, where it was further refined based on the results of the search. The full search strategy can be found in the Appendix A using keywords, MeSH terms, and Boolean operators. The selection of appropriate databases and application of the final search strategy for each database was guided by the lead supervisor to maximise the number of relevant articles located by the search. The search was undertaken in PubMed, ScienceDirect, Scopus, SPORTDiscus and Google Scholar, ending on 04 December 2024. A manual search for reference lists of relevant publications and using Google Scholar to scan gray literature to ensure that no relevant articles were excluded.

### 2.4. Study Selection

Duplicates and articles published before 1970 were removed, as collectively these databases report that they reliably index records from 1970 onward. The remaining title and abstracts were screened by the authors using the predetermined inclusion/exclusion criteria. Full texts were retrieved and evaluated against the inclusion/exclusion criteria independently, as summarized in Figure 1.

### 2.5. Data Charting and Data Items

Data was extracted into the data extraction table that was constructed collaboratively between the authors. Any disagreements were resolved. Data extracted included study design, participant characteristics, the intervention type, and the intervention delivery method. For this scoping review the dietary interventions were classified into major intervention approach (including supplements), altered nutrient intake (which included interventions where only one aspect of the diet was being modified), fasting/energy restriction, altered overall diet (such as changes in overall diet pattern, low carbohydrate diet) and any ‘other’ interventions could be classified into any of the aforementioned categories. Sleep outcomes were classified as sleep onset SOL, SE, sleep quality, waking after sleep onset (WASO), or TST. Methodological quality was assessed using NUQUEST. We abstracted data on article characteristics, participant athletic status, and nutrient intervention/exposure.

### 2.6. Critical Appraisal of Individual Sources of Evidence

#### 2.6.1. NUQUEST

Using NUQUEST for this scoping review involved systematically applying the tool to evaluate the quality and risk of bias (RoB) in each study included. The NUQUEST scoring system assessed RoB in nutrition studies by evaluating specific methodological domains critical to study quality [32]. These domains included participant selection, exposure and outcome measurement, confounding control, and reporting transparency. Each domain was rated as either “good”, “neutral”, or “poor”; domain ratings were reached using a prespecified point system already in place by the original researcher [32], reflecting the extent to which the study adheres to rigorous scientific standards. Higher scores indicated a lower RoB, while lower scores suggest greater susceptibility to methodological flaws. RoB tools for nutrition randomised controlled trials (RCTs), cohort, and case–control studies were utilised [32]. In this scoping review RCTs and cohort studies were applicable. A NUQUEST tool for cross-sectional studies does not currently exist; as such, the cohort worksheet was applied as the most appropriate available option because both designs are observational and assess exposures concerning outcomes without randomisation. Although cross-sectional studies measure exposure and outcome simultaneously, many risk-of-bias domains in the cohort tool (e.g., selection bias, measurement validity, confounding) are still relevant and allow for a structured and transparent appraisal. The authors independently performed RoB assessments for each included study, with all investigators participating in reviewing the assessments. Full NUQUEST assessments are attached in the Appendix A.

#### 2.6.2. The Paper to Podium Matrix

To enhance the practical applicability of the recommendations from this review, the checklist of criteria from the Close [33], paper, “From Paper to Podium (P2P): Quantifying the Translational Potential of Performance Nutrition Research,” was utilised. This framework serves as a critical tool for evaluating performance nutrition research and assessing its applicability in real-world practice. This operational framework assesses (1) research context, (2) participant characteristics, (3) research design, (4) dietary and exercise controls, (5) validity and reliability of exercise performance tests, (6) data analytics, (7) feasibility of application, (8) risk/reward and (9) timing of the intervention. A five-point numerical grading scale (−2, −1, 0, +1, +2) was used to calculate the total scores from each assessment area. This approach enhances clarity for practitioners by highlighting well-supported recommendations, enabling them to make informed decisions for the athletes they support. Details on the total scores for each study can be found in the Appendix A. The highest score possible on the P2P is +18, and the lowest score possible is −18. As alluded to in the methods and materials section, there is an operational framework for rating each study. Full P2P assessments are attached in ranking order in Appendix A.

### 2.7. Synthesis of Results

Charted data from included studies were synthesized using a descriptive analytical approach, consistent with scoping review methodology. Data extraction focused on key variables relevant to the review objective, including author, study design, population (elite athletes), sleep assessment method, nutrient intervention or exposure, timing of intervention, sleep outcome(s), and quality ratings (NUQUEST and Paper-to-Podium (P2P) scores). Studies were first categorised by design (e.g., randomized controlled trials, cohort studies) and athletic population characteristics. Interventions were grouped by type of nutritional strategy, such as macronutrient composition, micronutrient supplementation, or manipulation of meal timing. Sleep outcomes (e.g., sleep duration, quality, SOL) were classified according to the assessment method used (e.g., actigraphy, self-report, polysomnography). A tabular summary of the included studies was constructed to enable visual mapping of the evidence across the defined domains (Table 1). Frequency counts and narrative synthesis were used to identify common themes, gaps, and patterns in the evidence base. RoB and methodological quality were appraised using the NUQUEST tool, and translational potential was evaluated using the P2P framework. It is important to note that these tools were used in isolation from each other with NUQUEST demonstrating to academics how to assess the quality of nutrition research and P2P Matrix demonstrating to practitioners the translational aspect of applying this tool to their practice with elite athletes. Outcome values for both should be taken as separate entities that helped inform the overall synthesis of Table 2. These assessments were not used to exclude studies but informed the interpretation of findings in the context of practical application in elite sport.

## 3. Results

### 3.1. PRISMA Diagram

The search criteria strategy identified 1903 articles for screening shown in Figure 1. Following title and abstract screening, 70 studies were assessed for eligibility, from which 58 studies were excluded because they did not meet the inclusion/exclusion criteria. In total, 12 studies were included for the current review. Seven studies focused on dietary interventions reporting sleep outcomes, while five studies focused on sleep outcomes based on dietary exposure. Screening procedures are also outlined in Appendix A.

### 3.2. NUQUEST

NUQUEST classifications, as seen in the Appendix A categorised 10 of the 12 studies as good (+), and two were rated as neutral (0) regarding RoB.

### 3.3. Paper to Podium Matrix

Outcomes of the Paper to Podium Matrix using the included studies in the Appendix A.

### 3.4. Synthesis of Results

#### 3.4.1. Fruit

##### Kiwi Fruit

One included study investigated kiwi fruit [29]. Pittsburgh Sleep Quality Index (PSQI) global scores reduced significantly from baseline (6.47 ± 2.17) to post-intervention (4.13 ± 1.19; z = 91, *p* = 0.002) after consumption of 2 kiwis 1 h before bed for 4 weeks. Sleep quality improved significantly from baseline (1.53 ± 0.84) to post-intervention (0.27 ± 0.46; z = 78, *p* = 0.002). TST improved week to week from baseline to post-intervention (F (4, 44) = 6.653, *p* = 0.001, partial η2 = 0.38). TST increased from baseline 7.6 ± 0.75 h to 8.55 ± 0.44 h in week 4, a statistically significant increase of 0.83 ± 0.23 (*p* < 0.05). Number of awakenings reduced significantly from baseline to intervention: χ^2^ (4) = 12.6, *p* < 0.05. WASO reduced significantly from baseline to intervention: χ^2^ (4) = 12.5, *p* < 0.05. SE increased significantly from baseline to intervention: χ^2^ (4) = 21.2, *p* ≤ 0.001. statistically significant increase in SE compared to baseline in week 2 (*p* = 0.018), week 3 (*p* < 0.001), week 4 (*p* < 0.001), and week 5 (*p* < 0.001).

##### Tart Cherry Juice

One included study investigated tart cherry juice [38]. Significant interaction effects (group × time) between sleep quality variables and the consumption of tart cherry juice (TCJ) five times over 48 h. Specifically, TCJ intake influenced TTB (*p* = 0.015), WASO (*p* = 0.044), and Movement Index (MI) (*p* = 0.031). TTB showed significant differences (*p* = 0.005) before and after in the TCJ group and significant differences (*p* = 0.005) between the placebo group and the TCJ group after the intake of tart cherry juice. In the case of WASO, it showed significant differences (*p* = 0.024) before and after in the TCJ group. Levels of melatonin and cortisol did not change significantly.

#### 3.4.2. Dairy

One included study investigated the effect of dairy consumption on sleep [44]. It was indicated that no significant associations of sex with type of sport, subjective sleep quality, or frequency of milk or dairy product consumption were observed in 679 elite athletes. The men (n = 379) showed a lower frequency of dairy product consumption than the women (*p* < 0.001). There was also no statistically significant association between subjective sleep quality and frequency of milk consumption in all subjects and men; however, a greater frequency of milk consumption was significantly associated with good subjective sleep quality in women (*p* < 0.001). More frequent milk consumption was significantly associated with better subjective sleep quality during training periods in women only.

#### 3.4.3. Probiotic

One included study investigated probiotics [40] and found that muscle soreness was ~0.5 units lower (F (1, 343) = 42.646, *p* < 0.0001) and leg heaviness scores ~0.7 units lower (F (1, 344) = 28.990, *p* < 0.0001) in the probiotic group versus the placebo group in 19 elite rugby union male athletes over 17 weeks. Across both groups, as self-reported muscle soreness scores and salivary CRP concentrations increased, sleep quality, quantity, and motivation scores decreased. Conversely, as muscle soreness scores and CRP decreased, sleep quality, quantity, and motivation scores improved.

#### 3.4.4. Protein & Protein Derivatives

##### Protein & Tryptophan

Ferguson [34] demonstrated that the habitual sleep/wake behaviour during the 5-day pre-intervention period for bedtime, SOL, WASO, SE, wakeup time, and sleep duration on training days was 23:18 ± 1:12 (h:min) time, 18.8 ± 10.0 min, 52.6 ± 31 min, 90.0 ± 5.6%, 08:36 ± 01:18 (h:min), and 7.9 ± 1.1 h, respectively, and on non-training days was 22:54 ± 1:06 (h:min), 20.8 ± 13.1 min, 49.5 ± 23.6 min, 89.7 ± 4.3%, 07:18 ± 00:30 (h:min), and 7.1 ± 0.8 hr, respectively. The average habitual daily protein intake was 226.8 ± 53.5 g (2.6 ± 0.7 g·kg^−1^) on training days and 205.9 ± 64.8 g (2.4 ± 0.7 g·kg^−1^) on non-training days. No differences were observed for all sleep outcomes between consumption of the whey protein supplement (55 g), rich in tryptophan (1 g), and the placebo on training and non-training days.

##### Casein

Casein is a slow-digesting milk protein that provides a sustained release of amino acids, supporting muscle repair and overnight recovery [37]. Timing of casein (40 g) [37] before sleep or earlier in the day resulted in no significant time effect interaction on sleep scores [1,2,3,4,5,6,7] measured during the 6-day training period. The findings revealed a significant increase in training loads during the camp, leading to elevated indicators of fatigue and decreased performance between day 1 and day 6. Despite high protein intake (>2.5 g·kg^−1^ BM on average), no significant differences were observed between the protein supplement groups and the placebo in any of the outcomes. Macronutrient intake during the camp was similar among the groups for total energy, carbohydrate, and fat intake. However, protein intake was significantly higher in the pre-sleep PRO and Afternoon PRO groups compared to the Placebo group (*p* < 0.001). A significant time effect was noted for the total Hooper index score, fatigue, and DOMS, all increasing over time, except for sleep and stress.

##### A-Lactalbumin

α-Lactalbumin is a whey protein rich in tryptophan, linked to improved sleep quality and mood regulation [36]. Two studies included examined α-lactalbumin. MacInnis [36] investigated α-Lactalbumin and showed there were no differences between the collagen peptides (CP) and pre-sleep consumption of α-Lactalbumin in terms of TTB, TST or SE in 6 elite male endurance track cyclists. α-Lactalbumin: 568  ±  71 min, 503  ±  67 min, 88.3%  ±  3.4%; CP: 546  ±  30 min, 479  ±  35 min, 87.8%  ±  3.1%; *p*  = 0.41, *p*  =  0.32, *p*  =  0.74, respectively.

Gratwicke [39] demonstrated no effect of α-Lactalbumin on TST or WASO across the intervention period (*p* > 0.05). However, there was a significant condition-by-period interaction effect on SOL (*p* = 0.01). While similar at baseline and for the home game, SOL was significantly shorter in the α-Lactalbumin group for the no-game week and away game but remained stable in the placebo group.

#### 3.4.5. Energy Intake, Micronutrient Intake and Timing

##### Energy Intake

In the prospective cohort study by Condo [42], 32 female Australian Women’s Football League players showed no significant association between energy intake and any sleep outcome. For a 1 g increase in daily carbohydrate intake, WASO increased by 0.05 min (*p* = 0.010) and SE decreased by 0.01% (*p* = 0.007). For each 1 g·kg^−1^ increase in daily carbohydrate, WASO increased by 3.6 min (*p* = 0.007) and SE decreased by 0.6% (*p* = 0.007). For a 1 g increase in saturated fat intake, SOL decreased by 0.27 min (*p* = 0.030) and for a 1 g·kg^−1^ increase in saturated fat, SOL decreased by 17 min (*p* = 0.049). Condo [42] showed that for each 1 mg increase in daily iron intake, sleep duration increased by 0.55 min (*p* < 0.001) and SE increased by 0.05% (*p* < 0.001). For a 1 mg increase in sodium, sleep decreased by 0.012 min (*p* < 0.001). For each 1-μg increase in vitamin B12, WASO decreased by 1.7 min (*p* = 0.020) and SE increased by 0.4% (*p* = 0.033). For each 1-mg increase in zinc, SE increased by 0.23% (*p* = 0.006). For each 1-mg increase in vitamin E, SE decreased by 0.08% (*p* = 0.016). For each 1-mg increase in calcium, SOL reduced by 0.005 min (*p* = 0.015). For each 1 mg increase in magnesium, SOL was reduced by 0.02 min (*p* = 0.031).

In the other prospective cohort study, assessing energy intake, in 36 male elite athletes [43] every 1 g and 1 g·kg^−1^ increase in total daily protein intake was associated with a decrease in SE by 0.01% (*p* = 0.007) and 0.7% (*p* = 0.006), respectively. Every 1 Mega Joule (MJ) increase in total daily energy intake was associated with a three-minute increase in WASO (*p* = 0.032). Every 1 g and 1 g·kg^−1^ increase in total protein intake was associated with an increase in WASO by 0.04 min (2 s; *p* = 0.014) and 4 min (*p* = 0.013), respectively. Every 1 g and 1 g·kg^−1^ increase in evening sugar was associated with a decrease in TST by 0.1 min (6 s; *p* = 0.039) and 5 min (*p* = 0.027), respectively; an increase in SE by 0.002% (*p* = 0.015) and 0.2% (*p* = 0.021), respectively and a decrease in WASO by 0.012 min (1 s; *p* = 0.003) and one minute (*p* = 0.005), respectively. Every 1 MJ increase in evening energy intake was associated with an increase in SOL by 5 min (*p* = 0.011). Every 1 g and g·kg^−1^ increase in evening protein intake was associated with a decrease in SOL by 0.03 min (2 s; *p* = 0.013) and 2 min (*p* = 0.013), respectively. Every additional hour between the main evening meal and bedtime was associated with a decrease in TST by 8 min (*p* = 0.042) and a decrease in WASO by 2 min (*p* = 0.015). Every additional hour between the last food or drink (excluding water) consumed and bedtime was associated with a decrease in TST by 6 min (*p* = 0.014) [43].

In 115 elite athlete, Eroğlu [41] used RCSQ scores were used to compare the intake of nutrients in poor sleepers and good sleeper groups. The good sleeper group had higher energy (kcal), protein (g·kg^−1^ and g) and tryptophan (g·kg^−1^ and g) intakes (*p* < 0.05). Glycemic index (GI) values of the last meal were similar (*p* > 0.05).

##### Type of Macronutrient Pre-Sleep

In 14 Division 1 female soccer players, Greenwalt [35] concluded that sleep duration (*p* = 0.10, 0.69, 0.16, 0.17) and sleep disturbances (*p* = 0.42, 0.65, 0.81, 0.81) were not affected by high versus low kilocalorie (kcal), protein, fat, carbohydrate intake, respectively. Recovery (*p* = 0.81, 0.06, 0.81, 0.92) was also unaffected by high versus low kcal, protein, fat, or carbohydrate consumption. Consuming a small meal before bed may have no impact on sleep or recovery. Those who had more than 5 g of protein before bed had a recovery score of 11.41 percentage points lower than those who ate less than 5 g of protein pre-sleep; exact timing was not reported.

## 4. Discussion

The purpose of this scoping review was to map the current evidence and compare the nutrient strategies that potentially promote sleep in elite athletes. Through using both NUQUEST and the P2P matrix, this review has identified a table of evidence (Table 2) for practitioners to utilise, a scoring system that translates evidence to practice using the P2P matrix in Appendix A, and a demonstration of NUQUEST ratings to demonstrate methodological rigour in Appendix A. The 12 selected research papers provide valuable insights into the impact of various nutrient interventions and exposures and their effect on sleep outcomes in elite athletes and offer transparent directions for future research. They collectively contribute to the broader understanding of how nutrition and supplementation can influence elite athlete recovery and performance through sleep improvements and will be discussed in thematic order of intervention/exposure below.

### 4.1. Fruit, Dairy & Probiotics

#### 4.1.1. Fruit

Fruit-based interventions have demonstrated beneficial outcomes on sleep in elite athletes, with studies by Doherty [29] and Chung [38]. Kiwifruit consumption improved sleep quality, while TCJ intake enhanced several sleep metrics. Kiwifruit’s effects may be attributed to its melatonin [24 µg/g] and serotonin [5.8 µg/g] content, which support circadian rhythm regulation, along with its antioxidant and folate content, potentially reducing inflammation and insomnia [45]. Doherty [29] showed significant improvements in PSQI scores, SE, TST, and reduced number of awakenings, aligning with benefits seen in adults with sleep disturbances [45] and students with insomnia [46]. Similarly, Kanon [47] found that consuming fresh or dried green kiwifruit alongside an evening meal improved sleep and mood, possibly via increased serotonin metabolism, evidenced by elevated urinary 5-HIAA. Despite the open-label design, kiwifruit appears promising as a sleep aid, though further large-scale trials during competition phases are needed. Regarding TCJ, a randomised, double-blind, placebo-controlled study with 20 adults found that 7-day supplementation reduced napping time and increased SE, with significantly elevated melatonin levels (*p* < 0.05), though SOL and SE improvements were non-significant [48]. This suggests TCJ may enhance sleep duration via increased exogenous melatonin. Somewhat similar outcomes were established from this review, where Chung [38] found that five TCJ servings in 48 h resulted in improved sleep quality, despite non-significant changes in melatonin and cortisol. This may reflect increased antioxidant demands in elite athletes during training. While potentially effective, the high frequency of TCJ dosing may pose practical challenges for elite athletes in real-world settings and requires further investigation before this intervention is established as a standardised protocol in elite sport.

#### 4.1.2. Dairy

Yasuda [44] and Sato [49] both found consistent associations between habitual dairy consumption and improved sleep quality, particularly among women. Yasuda [44] conducted a cross-sectional study involving 679 Japanese elite athletes preparing for the 2016 Olympics. They found that higher frequencies of milk consumption (5–7 d.wk) were significantly associated with a lower risk of poor subjective sleep quality in female athletes. The association remained significant for women even after adjusting for confounders such as smoking, drinking, and sleep duration. These findings, consistent with large-scale adult data [49], suggest milk’s tryptophan content may contribute to improved sleep by promoting melatonin and serotonin production [50]. Female athletes can report different sleep experiences compared to males, potentially due to hormonal fluctuations (e.g., menstrual cycle effects) or differences in how they perceive and report sleep quality. This might make dietary interventions like dairy intake more detectably beneficial in women. While both studies are cross-sectional, they underscore a potential sex-specific benefit of dairy on sleep that warrants further controlled trials.

#### 4.1.3. Probiotics

Harnett [40] examined the effects of a 17-week multispecies probiotic on sleep and muscle soreness in elite rugby players. Improved self-reported sleep quality was observed, particularly as muscle soreness decreased. CRP levels, inversely associated with sleep quality, were less impactful in the probiotic group, suggesting reduced inflammation as a possible mechanism. Improved sleep quality correlated with better motivation and reduced muscle soreness in the probiotic group. This aligns with previous research showing that gut microbiota influences sleep quality by regulating neurotransmitters like serotonin, GABA, and melatonin [51,52]. GABA is primarily synthesized from glutamate through the action of glutamic acid decarboxylase, which requires pyridoxal phosphate (derived from vitamin B6) as a cofactor and acts as the principal inhibitory neurotransmitter in the central nervous system [51,52]. Comparable outcomes have been reported in non-athlete cohorts [53,54]. In conclusion, Harnett [40] suggests that long-term probiotic supplementation may enhance sleep quality in elite athletes; however, these results require replication across other elite athletic populations and contexts.

### 4.2. Macronutrients

#### 4.2.1. Protein & Derivatives

Studies investigating evening protein or tryptophan-rich supplements have yielded moderate-to-low benefits. Ferguson [34] found that evening whey protein supplementation did not enhance acute sleep duration or quality in elite male athletes. The findings contradicted the hypothesis that whey protein supplementation, rich in tryptophan, improved sleep outcomes due to increased tryptophan availability and melatonin synthesis. The study suggested that a ceiling effect may exist for sleep duration in populations already obtaining adequate sleep. Additionally, the high habitual protein intake could have limited the potential benefits of additional protein intake on sleep. A hypothesis for this ceiling effect, from a biological mechanism standpoint, is that tryptophan competes with other large neutral amino acids for transport across the blood–brain barrier via the large neutral amino acid transporter [55]. This competition influences tryptophan availability in the brain, subsequently affecting serotonin and melatonin synthesis, which are crucial for sleep regulation [55]. This hypothesis, despite lacking robust validity, is proposed as a potential limiting factor to the effectiveness of tryptophan interventions when elite athletes are already consuming high-protein diets [34,55]. This is particularly pertinent when an intake of protein sources high in large neutral amino acids and low in tryptophan reduces the ratio of tryptophan availability. This has, however, not been tested comprehensively in athletes [43].

Similarly, Valenzuela [37] found no effect of consuming protein pre-sleep on recovery. Sleep quality remained statistically unchanged throughout the study (*p* = 0.148), indicating that neither pre-sleep protein nor afternoon protein had a meaningful effect on sleep quality compared to the placebo. Additionally, the Recovery-Stress Questionnaire for Athletes (RESTQ-Sport) assessed stress and recovery markers, including sleep quality and disturbed breaks during sleep. These measures did not show significant differences between the groups. The lack of significant effects observed in this study could, again, be attributed to the already high dietary protein intake of cyclists, exceeding recommended levels.

Another comparable outcome was seen with 6 male elite athletes [36], showing no significant differences between the α-Lactalbumin and placebo (collagen peptides) groups in TTB, TST, or SE. Again, questioning the efficacy of a protein-specific, tryptophan-rich intervention when elite athletes already meet their nutrient targets. Gratwicke [50,56] did, however, highlight the efficacy of pre-sleep α-Lactalbumin consumption as a nutritional intervention to improve SOL in a semi-professional female team-sport cohort. Thus, α-Lactalbumin could potentially be utilised by female athletes to support sleep during a competitive season.

#### 4.2.2. Carbohydrate Intake Pre-Sleep

Evidence on pre-sleep carbohydrate intake in elite athletes remains inconsistent. High GI carbohydrates may promote sleep by triggering insulin release, which reduces competing amino acids in the blood and allows more tryptophan to enter the brain. This increased tryptophan availability can potentially enhance serotonin and melatonin production [50,56]. Falkenburg [43], Condo [42], Killer [57] and Greenwalt [35] reported variable impacts on SE and WASO, with high carbohydrate occasionally impairing sleep. Afaghi [50] found that the type and timing of carbohydrate may affect sleep, with improved SOL from a high-GI meal seen 4 h pre-bed. Contrastingly Daniel [58] suggested that overall diet quality may be more influential than isolated carbohydrate timing. These findings support that energy balance and nutrient quality should be prioritised over carbohydrate manipulation.

#### 4.2.3. Energy Availability and Meal Timing

Falkenberg [43] reported that a longer interval between the last meal and bedtime reduced TST, while eating too close to bedtime increased WASO. Evening protein intake was associated with SOL, yet paradoxically, overall higher protein and sugar intake correlated with poorer sleep quality. Eroğlu [41] concluded that good sleepers consume more total energy, protein, and tryptophan than poor sleepers. This again suggests that specific interventions may have limited utility when nutritional requirements are already met. Greenwalt [35] further confirmed that macronutrient timing did significantly impact sleep metrics, underscoring the complexity of dietary effects on sleep in elite athletes.

### 4.3. Micronutrients

Micronutrients such as vitamin B6, B12, zinc, iron, and magnesium serve as essential cofactors for a range of metabolic enzymes implicated in sleep regulation and neurotransmitter synthesis [59]. Vitamin B6 acts as a coenzyme in the conversion of tryptophan to serotonin. Meanwhile, GABA is either reabsorbed by specialized transporters in neurons and glial cells, or it is broken down by the enzyme GABA transaminase [59]. Vitamin B12 supports melatonin synthesis and regulates the circadian rhythm, and zinc may directly modulate GABAergic activity and melatonin synthesis [59,60,61]. Magnesium is important for both melatonin production and modulation of GABAergic neurotransmission [62,63,64]. The key roles of these micronutrients as cofactors in enzymatic reactions underpin their importance in supporting healthy sleep physiology in elite athletes, as seen by Condo [42], where a positive correlation was established between SE and intake of vitamin B12, zinc, and iron. A study using ZMA (zinc, magnesium, B6) also showed improvements in SOL and SE [59]. Ensuring adequate intake of these micronutrients, especially in elite female athletes at risk of deficiencies, remains a foundational component of sleep-supporting nutrition.

### 4.4. Limitations & Future Directions

A limited sample size of randomised controlled trials and exposure studies exists investigating nutrition strategies to promote sleep in elite athletes. The small sample size, in this scoping review (n = 12), may exclude rigorous studies undertaken in highly trained athletes or athletes in lower category demographics to provide preliminary evidence. Therefore, given that these were excluded, single study reference points were used for kiwi fruit and tart cherry consumption. Further research is required to confirm their efficacy and explore their role in improving sleep among other elite athletic populations. Using similar intervention protocols from both trials with larger cohorts (n = 50), over sustained periods of intense training or game (4+ weeks) in both males and females would help to standardize the protocols implemented. No studies met the criteria for carbohydrate intake pre-bed in elite athletes despite previous evidence in non-athlete cohorts. The assumption that high-GI carbohydrate improves sleep quality needs to be evaluated with original research. Future research should focus on large-scale, well-controlled trials investigating the efficacy and practical implementation of nutrient timing and specific food-based interventions, such as fruit and probiotic supplementation, to potentially promote sleep quality in diverse elite athlete populations during competition phases.

## 5. Conclusions

The 12 studies reviewed provide valuable insights into the potential impact of nutrient strategies to promote sleep in elite athletes. In summary, promising interventions such as kiwifruit, tart cherry juice, and dairy consumption were identified as having high potential in elite athletes, despite single study reference points. Dairy and milk consumption improving sleep was limited to female athletes only, so further research examining higher doses in different ethnicities is plausible. Also, given the proposed biological mechanisms of sleep improvement by consuming high-GI carbohydrate pre-sleep, varying GI food types should be tested on large sample sizes and offer further clarity on the carbohydrate debate. Given these inconsistent results, individual nutrition recommendations to simply meet energy demands are recommended to maintain sleep quality. Tryptophan-rich supplementation may hold promise, but a ceiling effect may exist in elite athletes consuming >2.5 g·kg^−1^ BM, when tryptophan is competing with other large neutral amino acids. Extended probiotic supplementation may result in improved sleep scores as a byproduct of reduced muscle soreness.

This scoping review provides a template for practitioners working with elite athletes and uses both the P2P Matrix [33] and NUQUEST [32] to offer additional clarity in the practical application of any strategies utilized while accounting for risk of bias across studies. Ultimately, advancing the science of performance nutrition requires translating evidence into context-specific strategies, rather than generalising results from non-elite cohorts. This will help to ensure that the guidance for elite athletes is grounded in research that reflects their unique demands, environments, and recovery needs.

## Figures and Tables

**Figure 1 sports-13-00342-f001:**
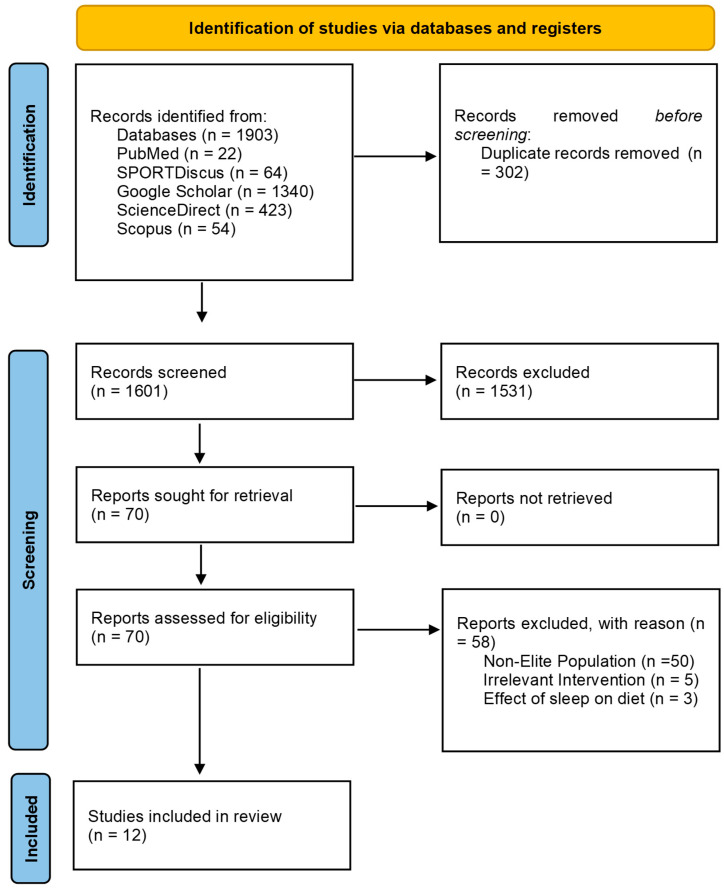
PRISMA Flow Diagram for nutrition interventions and exposure to promote sleep in elite athletes.

**Table 1 sports-13-00342-t001:** Nutrient strategies on sleep outcomes. Studies ranked from most to least translational to practice, using the P2P ratings.

Author	Study Design	Population	Sleep Assessment	Nutrient Strategy	Timing	Sleep Outcome	NUQUEST Rating	P2P Rating
**Ferguson et al., 2022** [34]	Double-blinded, counterbalanced, randomised, crossover study	15 elite male AFL players	Wrist activity monitors and sleep diaries	55 g whey protein (1 g tryptophan)	Evening time (on training and 2 non-training days) during pre-season.	No improvement in sleep duration or quality who already have adequate sleep	Good	14
**Greenwalt et al., 2023** [35]	Retrospective Study	14 Div 1 female soccer players. Menstrual cycle not tracked	WHOOP bands (24 h monitoring), surveys	Pre-sleep protein intake (>5 g)	Various times during the in-season	>5 g protein before bed lowered recovery score by 11.41 percentage points, but no significant effect	Neutral	13
**MacInnis et al., 2020** [36]	Randomised, Double-Blind Cross-Over Design	6 male elite cyclists	Wrist-based actigraphy	A 40-g serving of α-Lactalbumin with essential amino acids (19.7 g), leucine (4.3 g], and tryptophan (1.9 g)	2 h pre-sleep (3 consecutive evenings) during an in-season training camp	No improvement in TST, time in bed, or SE	Good	13
**Valenzuela et al., 2023** [37]	Double-blinded, parallel-group, three-arm, randomised controlled design.	24 Pro U23 Cyclists. Gender not specified. Menstrual cycle not tracked.	Self-rated sleep quality	40 g casein	40 g casein (10.30 PM), 40 g casein (6.30 PM), or 40 g carbs (10.30 PM) at the 6-day training camp in pre-season.	No significant group differences in sleep quality	Good	13
**Chung et al., 2022** [38]	Double Blind Randomised Controlled Trial	19 elite female hockey players. None experienced menstruation during the study.	Sleep quality, melatonin & cortisol levels	30 mL tart cherry juice in 200 mL water	5 doses over 48 h (morning and evening) during camp training in pre-season	No effect on melatonin/cortisol, but improved sleep quality	Good	12
**Gratwicke et al., 2023** [39]	Double-blind randomised placebo-controlled trial	18 semi-professional female rugby union players. Menstrual cycle not tracked	Actiwatch 2, Philips Respironics, Pennsylvania, USA	40 g of α-Lactalbumin protein powder (4.8 tryptophan/100 g]	300 mL of water and consumed approximately 2 h before going to bed in pre-season and in-season in 4 × 7 day blocks	Pre-sleep α-Lactalbumin consumption improved SOL in a semi-professional female team-sport cohort	Good	12
**Harnett et al., 2021** [40]	Double Blind randomised control trial	19 elite male rugby athletes	Self-reported sleep quality, salivary biomarkers	Daily probiotic (Ultrabiotic 60 TM & SBFloractvi TM with 250 mg Saccharomyces boulardii) vs. placebo	Daily for 17 weeks in-season	Across both groups, increased muscle soreness & C-Reactive Protein (CRP) reduced sleep quality; decreased soreness & CRP improved sleep	Good	12
**Doherty et al., 2023** [29]	Open Label Trial	15 elite national sailors and middle-distance runners. Menstrual cycle was not tracked in females.	Questionnaire battery (RESTq Sport *, PSQI **, CSD-C ***, RU- Sated, sleep diary)	130 g or 2 kiwi fruit. Following the baseline assessment (Week 1) all subjects began the intervention (Weeks 2–5).	1 h pre-bed during pre-season (n = 10) and in-season (n = 5)	Improved sleep quality, increased TST & SE%, reduced number of awakenings and reduced WASO	Good	11
**Ergolu et al., 2024** [41]	A Pilot Study: Cross-sectional design	115 elite athletes (swimming, canoeing, archery, volleyball, taekwondo)	Richard-Campbell Sleep Scale (RCSQ)	24-h food consumption recorded and analysed using nutritional software (Nutrition Information Systems BeBiS version 8.1)	No specific intervention and 90% participants in-season	Insufficient calorie intake is associated with poorer sleep. Good sleepers consumed 1.6 g/kg BM protein and 1350 mg tryptophan	Good	11
**Condo et al., 2022** [42]	Prospective Cohort Study	32 elite female Australian football league players. Menstrual cycle not tracked.	Activity monitors, sleep diaries	Diet monitored (carbohydrate, fat, iron, zinc, B12)	10 consecutive days during pre-season	Increased carbohydrate intake is associated with increased WASO & decreased SE. Higher iron, zinc, and B12 improved sleep outcomes	Good	10
**Falkenberg et al., 2021** [43]	Prospective cohort study design	36 male elite AFL	Wrist actigraphy, sleep diaries	Evening sugar, protein and meal timing analysis	10 consecutive days during pre-season	High sugar and a long time between eating and bed reduced TST; evening protein reduced SOL	Good	10
**Yasuda et al., 2019** [44]	Cross-sectional Study	679 Japanese elite athletes (2016 Rio candidates] with menstrual cycle not tracked in females).	Self-reported questionnaires	Frequency of milk/dairy consumption (0–2x, 3–5x, 6–7x per week)	Overall daily dairy intake in pre-season before the 2016 Olympic Games	Higher milk consumption is associated with better sleep quality in women, not men	Neutral	10

* Recovery-Stress Questionnaire for Athletes (RESTq Sport), ** Pittsburgh Sleep Quality Index (PSQI) *** Consensus sleep diary (CSD-C), Total Sleep Time (TST), Sleep efficiency (SE), Sleep Onset Latency (SOL), wakenings after sleep onset (WASO).

**Table 2 sports-13-00342-t002:** Nutrition Strategies to Promote Sleep in Elite Athletes.

Nutrition Strategy	Implementation	Sex	Sport	Time of the Season	Efficacy of Strategy *
**Kiwi fruit**	2 x kiwifruit (1 h Before bed) may improve sleep duration, SE, number of awakenings, WASO, sleep quality, and reduce fatigue the morning after.	Male & Female	Sailing & Endurance	In-season and pre-season	High Potential
**Tart Cherry Juice**	30 mL tart cherry juice consumed 5 times over 48 h, morning and evening, may improve TTB, reduce periods of WASO, and reduce movement index while sleeping.	Female	Hockey	Pre-season	High Potential
**Milk & Dairy consumption**	Greater frequency of milk consumption (5–7 d/wk.) is associated with a reduced risk of deteriorating perceived sleep quality	Female	Multiple	Pre-season	High Potential
**Protein & Tryptophan**	Higher evening protein intake (>1 h pre-sleep) is associated with shorter SOL. Good sleepers tend to consume protein of ~1.6 g·kg^−1^ per day and a tryptophan intake of ~1350 mg. Improvements inconclusive in those already consuming a high-protein diet (>2.5 g·kg^−1^ BM).	Male & Female	AFL, Cycling, and Rugby Union	In–season and pre-season	Moderate Potential
**A-lactalbumin**	No difference in sleep quality compared to collagen peptides, but it has been shown to potentially improve SOL during a season.	Female	Rugby Union	In-season and pre-season	Moderate Potential
**Probiotic**	Daily probiotic (Ultrabiotic 60 TM & SBFloractivit TM with 250 mg Saccharomyces boulardii) consumption for 17 weeks may help leg heaviness and muscle soreness with reduced CRP concentrations & muscle soreness being related to improved sleep quality and quantity.	Male	Rugby Union	In-season	Moderate Potential
**Zinc, Iron & B12**	Optimal levels of Zinc, Vitamin B12, and Iron may improve sleep quality.	Female	Australian Football League	Pre-season	Moderate Potential
**Timing**	Every additional 1 h between the main evening meal and bedtime may be associated with a decrease in TST and reduced periods of WASO.	Male	Australian Football League	In-season	Moderate Potential
**Total Calorie Intake**	Insufficient calorie intake is associated with poorer sleep, so appropriate refuelling strategies to meet caloric demands may increase TST.	Male & Female	Swimming, Canoeing, Archery, Volleyball, Taekwondo	In-season & pre-season	Moderate Potential
**Carbohydrate intake pre-bed**	Carbohydrate intake to meet energy & recovery needs should be well planned, as high carbohydrate intakes and high sugar intake pre-bed can be associated with increased WASO and reduced SE.	Male & Female	Multiple	In-season & pre-season	Uncertain Potential (due to conflicting evidence)

* Traffic Light System to demonstrate High, Moderate and Uncertain Potential.

## Data Availability

The original contributions presented in this study are included in the article/Appendix A. Further inquiries can be directed to the corresponding author.

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
