# Peer review of "Nutrition Strategies to Promote Sleep in Elite Athletes: A Scoping Review"

_sports, 2025, doi:10.3390/sports13100342_

Round 1
Reviewer 1 Report
Comments and Suggestions for Authors
This scoping review synthesizes evidence on nutrition interventions and exposures aimed at improving sleep in elite athletes. Following the PRISMA guidelines, the NUQUEST tool for bias assessment, and the P2P Matrix for practical applicability, the authors reviewed 12 studies. Key findings indicate that kiwifruit, tart cherry juice, and dairy show promise for enhancing sleep quality and duration. Meanwhile, protein-based interventions reach a ceiling effect in athletes already consuming more than 2.5 g/kg of protein. The review emphasizes sex-specific responses, with dairy benefits observed only in females, and identifies gaps in evidence regarding carbohydrate timing. However, major revisions are required before publication.
Exclusion of Adolescent Athletes: Eligibility criteria (≥18 years) overlook elite groups (e.g., Olympians) that include teens. This restricts generalizability and does not address sleep needs in younger athletes.
NUQUEST lacks a tool for cross-sectional studies (used in 3/12 studies), yet these were rated Good/Neutral without justification.
Dairy Effects: Abstract/Conclusions claim that dairy consumption improves sleep, but results show benefits only in females.
Table 5 ranks interventions by evidence strength.
A-lactalbumin appears throughout (Abstract, Table 5, Results, Discussion); it should be corrected to α-lactalbumin. Furthermore, the abbreviation (LA) should be specified as α-lactalbumin on its first mention (abstract).
Authors should fix formatting errors in reference citations according to the journal instructions. Some formatting reference [x] and (x). furthermore, [x] not superscript
Line 268 and table 5 catergorised should be categorised
The unit is 2.5g kg.BM should be 2.5g kg BM in the abstract and conclusion
What is the difference between kg and kg-1 (some places use superscripts, others don't.)
All abbreviations should be mentioned on first use, for example, NoA
Database names Science Direct should be (ScienceDirect) and SportsDiscus should be (SPORTDiscus).
The grammar and English throughout the manuscript require minor revision.
Comments on the Quality of English LanguageThe grammar and English throughout the manuscript require minor revision.
Author Response
All comments and responses are attached in the word document.

Reviewer 2 Report
Comments and Suggestions for Authors
This article presents a scoping review of nutritional strategies with the objective of improving sleep in elite athletes. The authors employ a robust dual-methodological approach, utilising the NUQUEST tool to assess risk of bias in nutritional studies and the Paper-to-Podium (P2P) Matrix to evaluate the practical relevance of findings. A total of twelve studies were included in the review, with the most promising interventions being kiwifruit, tart cherry juice, and dairy consumption, especially in female athletes.
The following section will address the limitations and critiques of the present study.
- The present study is informed by a modest body of evidence, as indicated by the sample size of 12 studies.
Despite the implementation of extensive database searches, a mere 12 studies were found to be in accordance with the stipulated inclusion criteria. This limitation results in a reduction in the study's power and its ability to generalise the findings.
It is important to note that some conclusions are based on single studies, for example, those on kiwifruit. This means that these conclusions must be interpreted with caution.
- The present analysis suggests the possibility of overinterpretation.
It is evident that several findings of a modest nature have been designated as "promising", despite the absence of a substantial effect on objective sleep outcomes.
The hypothesis of a protein/tryptophan "ceiling effect" in elite athletes is not only plausible but also worthy of further investigation. However, at present there is a lack of robust validation for this hypothesis.
- The analysis has been subject to a limited consideration of confounders.
It is evident that several significant factors have not been given due consideration or scrutiny. These include training fatigue, the menstrual cycle and the competition phase.
Suggestions for Further Development
It is imperative to elucidate the underlying biological mechanisms, including but not limited to amino acid competition, GABA, and micronutrient cofactors.
It is imperative that recommendations are stratified by sport type, competition period, sex, and stress level.
It is recommended that a standardized trial protocol be proposed for future interventional studies.
It is recommended that practitioners be provided with visual summary tools, for example in the form of a decision tree or infographic.
The following observations are hereby made:
This text constitutes the final evaluation.
This is a methodologically robust and practically valuable review with a clearly defined scope. Nevertheless, the evidence base upon which it is founded is limited in scope and heterogeneity, frequently consisting of small or single studies. While it offers actionable insights for sports dietitians and applied scientists, it highlights the urgent need for larger-scale, well-controlled intervention trials in real-world elite athletic contexts.
Author Response
All comments and responses are within the word document.

Reviewer 3 Report
Comments and Suggestions for Authors
I appreciate the opportunity to review this very interesting and important article. This was a well conducted and comprehensive systematic review; the steps and procedures of the scoping review were specified well and rigorous. Having done several of these as well, I was satisfied with the fact that all of the steps were carried out. I found that the conclusions/results of value which provide good take home messages. I do have some minor concerns about flow/sentence structure.
- Line 59 – I don’t think it is necessary to say “mild cognitive impairment” because “cognition” is mentioned earlier. The sentence is structured such that it means this this “negatively impacts”….”cognition” but it is not necessary to say it negative impact mild cognitive impairment. So if it impacts mild cognitive impairment, does that then mean that there is less mild cognitive impairment?
- Line 61 – This should be “sports” plural.
- TTB and other acronyms are bewildering at times making it very difficult to read. Please consider reducing the number of acronyms; it was frustrating to read at times.
- Line 102 – “elite” should be followed by “athletes”
- Line 116 – “Therefore, a scoping review….”
- Table 3 – I was not following the order of the studies listed. They are not in alphabetical order by first author’s last name or by year of publication. It does not appear to be grouped by intervention.
- Line 289 – I was getting lost on the acronyms. What is NoA?
- Line 318 – This sentence is difficult to interpret with the double negative.
- In the 3.4.4.1 section, it might be helpful to describe in the first sentence what the intervention is. Many of us may not know what Casein is, A-lactalbumin, etc. This is a consideration for the other interventions outside of this section. Many of us many not know specifically what is meant by a particular intervention.
- What is PLA and AFWL?
- What is a “bye week” – Is this like a home game?
- Loved Table 5
Author Response

(The authors gave the same response as above.)

Reviewer 4 Report
Comments and Suggestions for Authors
OVERVIEW
This scoping review has many flaws. The problems prevent it from justified, replicable, sufficiently transparent or having reliable conclusions.
There is some good practice here which I have listed, too. The standard of written English is fine, and there is good transparency in places.
Given there is no consensus on what a scoping review is, it is hard to judge what this article should achieve. Most scoping reviews focus on what is possible to evaluate, either because of quantity of evidence (the most studies of similar design), or quality of evidence. Quality has a large number of possible indicators, including study design (RCTs obviously best, Hierarchy of Evidence is relevant), size of participant cohort, generalisability of their population, consistency of exposure and measurement methods, but also endless other possibilities. It appears that the P2P framework doesn’t prioritise RCTs which suggests it isn’t a scientifically robust framework for decision-making, merely a suggestive one. According to authors, all of these studies scored relatively well on NUQUEST which to me suggests authors were not sufficiently critical in applying that checklist, too.
At any rate, this scoping review is focused on recommending which interventions would be most promising to try in future research because they seem to have had the most benefits so far and had high scores on the combined checklists. This scoping review ends up arguing that the interventions likely to produce sleep benefits in elite athletes are kiwi fruit, based on a pre-post study design on 15 runners, or tart cherry juice (19 participants, RCT, 48 hours or monitoring), or milk/dairy consumption (mostly based on a single large cross sectional self-report study). None of these studies are RCTs over a decently long time period. Two are tiny studies and pre-post designs are especially biased. None of them are convincing as basis for promising interventions. The authors say that there is no NUQUEST checklist for cross-sectional studies but don’t explain that the cohort checklist was applied to the cross-sectional study design to produce NUQUEST scores for the Xsect studies (information only made clear by looking in the supplemental files)… why would this be ok? Critical appraisal checklists are very customised to study design but also they are not comparable between study designs: a high score for a cohort study would not mean that cohort study had evidence as robust as an RCT with same score. So to pool the studies in one group, to apply a checklist to an ineligible study design, to treat the pre-post or cross-sectional studies as equal or ‘better’ evidence of intervention benefits, due to equal/higher score on their checklist than the RCTs… that isn’t robust practice.
MAJOR issues
The abstract isn’t clear enough about methods or results. I would like the abstract to say that the median participant group size was 19 (ie, small), that many study designs were considered together, and something about fact (how?) that NUQEST and P2P were combined to produce estimates of what might be “strongest evidence” for planning future research studies.
In methods, It is not clear how NUQUEST and P2P were combined. It seems like thresholds were identified that were derived from combinations of NUQ & P2P scoring matrices, but this is merely alluded to as “novel application” without obvious justification for the combined thresholds used or statements what they were.
Tables 1, 2 and 4 are unnecessary. Table 4 is useful (good for transparency) but could go in supplemental. Table 1 doesn’t seem to be part of the evaluation. The colours in Table 2 don’t match the colours used in Table 3, and given the colours are all same in Table 3… just not necessary to state the Table 2 info in a table format.
MINOR issues
(Since no definition what a scoping review is or what it tries to achieve): last sentence in Abstract in Methods could say, instead of “research merited”: this scoping review tried to identify the interventions that could be best to research in future. Say what this scoping review wanted to achieve.
References are supplied for definition of “elite athlete” but a sentence or 2 to explain the definition used by authors in this review would help reader in Methods.
Statement of who did which work using initials (section 2.4) rather than mention of student/supervisor would be most appropriate; readers shouldn’t care who was student/supervisor.
A list of abbreviations & their meanings at the end would be helpful. Abbreviations used in each table should be listed in table notes to help reader not have to dig back thru paper to find out what abbreviations are.
Tables across multiple pages should have table headers repeated.
There’s a comment in Table 5 about “Literature mixed on high GI carb pre-sleep for elite athletes” : this seems to be a comment that belongs in Discussion and Not in a Results table where the table is summary of analysis of primary single research studies.
OTHER comments
The numbers in Figure 1 add up.
The search strategy is transparent and reasonable.
Something wrong with my copy of the article, the pages after 12 had “page 23 of 23” on every page in the header… this is minor but would merit fixing.
I don’t understand how milk products would tend to be well-tolerated in Japanese populations (Yasuda study); aren’t most adult Japanese lactose intolerant? Aren’t most the world’s adult population lactose or casein intolerant?
Why would the small amounts of melatonin in Kiwi fruit or tart cherry juice be ideal dosage compared to a pill/capsule supplement?
Ditto for tryptophane and some of the other potentially biologically active nutrients being tested.
Author Response
Thank you for your comprehensive assessment of our manuscript. Significant changes have been made to the manuscript with all comments and responses attached in the word document.

Round 2
Reviewer 1 Report
Comments and Suggestions for Authors
The authors have taken into consideration the most comments/suggestions of the reviewers during the revision of the manuscript.
Author Response
Thank you for reviewing our paper and, as you mentioned, the authors took all your comments into consideration when altering the manuscript. It is very much appreciated.